# Applications of *Ulva* Biomass and Strategies to Improve Its Yield and Composition: A Perspective for *Ulva* Aquaculture

**DOI:** 10.3390/biology11111593

**Published:** 2022-10-30

**Authors:** Clara Simon, Marcus McHale, Ronan Sulpice

**Affiliations:** Plant Systems Biology Laboratory, Ryan Institute & Marei Centre for Marine, Climate and Energy, School of Biological & Chemical Sciences, University of Galway, H91 TK33 Galway, Ireland

**Keywords:** *Ulva*, metabolites, aquaculture, bioremediation, microbiome, strain-selection, environmental factors

## Abstract

**Simple Summary:**

Green sea lettuce (*Ulva* spp.), with its worldwide distribution and remarkable ability to grow rapidly under a range of conditions, represents an important natural resource that is still under-exploited. Its biomass can be used for various applications in the food, feed, pharmaceutical, nutraceutical, biofuel and bioremediation sectors. However, knowledge of *Ulva* genetics, its environmental responses and microbial interactions are far from complete. This knowledge gap is a major constraint for the development of *Ulva* aquaculture and further investigation of these factors is needed to advance strain selection for yield and biochemical composition in a broad range of cultivation environments. In this review, after presenting the characteristics of the biochemical content and the recognised applications of *Ulva* biomass, we present the established knowledge and highlight areas requiring greater investment to develop a sustainable and profitable *Ulva* aquaculture industry.

**Abstract:**

Sea lettuce (*Ulva* spp.), with its worldwide distribution and remarkable ability to grow rapidly under various conditions, represents an important natural resource that is still under-exploited. Its biomass can be used for a wide range of applications in the food/feed, pharmaceutical, nutraceutical, biofuel, and bioremediation industries. However, knowledge of the factors affecting *Ulva* biomass yield and composition is far from complete. Indeed, the respective contributions of the microbiome, natural genetic variation in *Ulva* species, environmental conditions and importantly, the interactions between these three factors on the *Ulva* biomass, have been only partially elucidated. Further investigation is important for the implementation of large-scale *Ulva* aquaculture, which requires stable and controlled biomass composition and yields. In this review, we document *Ulva* biomass composition, describe the uses of *Ulva* biomass and we propose different strategies for developing a sustainable and profitable *Ulva* aquaculture industry.

## 1. Introduction

Species of green macroalgae from the genus *Ulva* (Phylum Chlorophyta, Class Ulvophyceae, Order Ulvales, Family Ulvaceae) are among the most abundant seaweed species, being omnipresent in coastal communities around the world [1]. *Ulva* comprises diverse species which present two main morphologies, either tubular monostromatic (single cell layer) or foliose distromatic (two cell layers) [2,3]. Some species also present both morphotypes, such as *U. compressa* [4,5]. Thus, this morphological diversity is not solely explained by genetic variability, but also by considerable morphological plasticity in response to environmental conditions and variations in the associated microbiome [1,6,7]. For this reason, species identification based on morphological characters is largely unreliable and genetic information is necessary to properly identify *Ulva* species [8,9,10]. Currently, among the 400 *Ulva* species described, only ca. 40 species have been recognised taxonomically via the use of genetic information [11,12].

From an economic point of view, *Ulva* spp. biomass has long been recognised as sustainable and valuable. It contains valuable metabolites, including bioactive compounds, which can be used in the food, pharmaceutical, nutraceutical, or biorefinery industries [13,14]. Moreover, high growth rates, as well as high rates of carbon, nitrogen and phosphorus assimilation and the ability to adapt rapidly to variable environmental conditions, make *Ulva* spp. “physiologically robust” organisms and suitable candidates to be used as a biological filter (macroalgal “nutrient traps”) to mitigate eutrophication [15,16]. *Ulva* species have already proven their effectiveness for nutrient removal from water, e.g., in finfish and shellfish aquaculture, bio-reactor sludge, or industrial wastewater treatment [17,18,19,20,21,22]. However, there are still large knowledge gaps around their efficiency for nutrient removal, especially in “extreme environmental conditions”, such as low salinity, high nutrient loads, and varying temperatures. We can expect that some *Ulva* species or even strains are more suitable than others for nutrient removal in given environmental conditions [23] and identifying these might allow increases in *Ulva* bioremediation efficiency. Similarly, in addition to variation in nutrient removal capacity, species and strain-based variation in biomass composition are observed [23] and it is of importance for the valorisation of *Ulva* biomass. Thus, a strain selection programme tailored to both bioremediation efficiency and use of the accumulated biomass could lead to significant gains in yield and economical value of *Ulva* aquaculture, while reducing eutrophication. 

Therefore, due to the environmental impact of *Ulva* spp., but also their potential industrial applications, more research effort should be invested to document the natural variation present in the growth rates of the *Ulva* genus in different environments and how this relates to biomass composition. In this review, we summarise the current knowledge of *Ulva* biochemical composition, list the current uses of its biomass and comment on the different strategies we foresee in developing *Ulva* cultivation and the industrial applications of its biomass. 

## 2. *Ulva* Biomass Composition and Its Potential Applications

Green seaweeds of the genus *Ulva* have been intensively studied because of their accessibility and abundance in the intertidal zone of many oceans worldwide, as well as for their valuable chemical content. which makes them a target for a number of economically attractive industrial applications [24,25]. *Ulva* spp. biomass contains high levels of nutritional elements, such as proteins, carbohydrates, polysaccharides, minerals and lipids [26,27,28], which compares favourably with major land crops, especially for minerals (Table 1). Interestingly, the relative abundance of these compounds varies according to genetic differences between species and populations as well as environmental conditions, such as temperature, salinity, irradiance and nutrient composition of the water [25,29,30,31,32,33,34,35].

*Ulva* protein content is highly variable, from 9 to 29% of the dry weight (DW) [46]. The highest protein content, 29% (DW), was recorded in *U. lactuca* collected from North Yorkshire in the United Kingdom [47] and the lowest protein content was reported in the same *Ulva* species collected in Tunisia during the summer [46,48], which suggests a role of temperature in regulating protein content, similar to land plants [49,50]. Aspartic and glutamic acids are the most abundant amino acids and taken together can represent up to 26 and 32% of the total amino acids in *U. rigida* and *U. rotundata*, respectively [51,52]. Other essential amino acids required for animal and human nutrition are also abundant in *Ulva* (Table 2), with valine, leucine, lysine and threonine together representing 42% of the total amino acid content in *U. lactuca* [48]. 

*Ulva* is also considered an important source of minerals, with a relatively high ash content of 14% to 52% DW depending on the species and the growth conditions [26,53,54,55]. Potassium, magnesium, sodium and calcium are the main minerals in *Ulva* biomass (Table 1) [26,56].

*Ulva* spp. biomass is relatively low in energy due to a low lipid content (1 to 12% DW) and high carbohydrate content (41 to 50% DW) [26,57,58]. The carbohydrates are classified into three major groups, the water-soluble ulvans (8–29% DW) [59,60], the structural water-insoluble cellulose (9% DW) [61], and the non-structural water-insoluble starch (10% DW) [23]. Ulvans are mainly composed of sulfated rhamnose, uronic acids (glucuronic acid and iduronic acid) and xylose and are the major polysaccharides present in the cell wall [62]. The high carbohydrate content and the presence of valuable polysaccharides make *Ulva* spp. of interest for the production of pharmaceutical products and biofuels.

Minerals, proteins and fibres (Ulvans and cellulose) make *Ulva* biomass suitable for food and feed industries. In Asian countries, *Ulva* is popularly consumed as a foodstuff or can be used as an extract, for example, in health supplements [65]. In Western countries, seaweeds are mostly used as food additives or extracts and the use of *Ulva* as a foodstuff is still marginal [66,67]. This low popularity as a foodstuff might be partly explained by the legislation in place in Europe. Indeed, since 1997, according to the online European Novel Food Catalogue, only one *Ulva* species, *U. lactuca*, is allowed for human food consumption in all European countries, being classified as “non-novel food”. An exception to this is in France, where all *Ulva* species are accepted as food [13,68]. Molecular improvements to species classification, as well as sequencing type specimens in herbariums, have now determined that the species *U. lactuca* is absent in the East North Atlantic Area [11,69]. With this finding, it is highly likely that much of the *Ulva* spp. biomass consumed in Europe is misattributed and a change in legislation may be necessary. Including other *Ulva* species in the European list of species suitable for human consumption would encourage the consumption of endemic seaweed species and may contribute to the expansion of this market. 

However, special attention should be paid to the use of *Ulva* spp. as food for humans regarding food safety regulations [25]. While *Ulva* spp. can be natural accumulators of beneficial compounds, they can also accumulate toxic elements, such as heavy metals (e.g., mercury, arsenic, lead and cadmium) [70,71,72]. In Europe, there are strict legal limits concerning the maximum safe exposure levels for heavy metals. These limits are based on the recommendations of the Joint FAO/WHO Expert Committee on Food Additives (JECFA) and the European Food Safety Authority’s Expert Group on Contaminants in the Food Chain (CONTAM Group) and are established in legislation by Commission Regulation (EC) No. 1881/2006 [73].

In the 19th and early 20th centuries, seaweeds and in particular, the *Ulva* spp. were commonly used as feed for cattle, horses, and poultry in Europe (Norway, Scotland, France) and America [74]. However, there are limited data on the digestibility and energy values of *Ulva* for animals and especially for ruminants. This would depend on the biochemical content of the seaweed and also on the adaptation of the animal to this particular feed [53,75]. Their use as substitutes for land-based products for protein and other essential nutrients, such as minerals and vitamins has already been tested and led to *Ulva* biomass inclusion as feed for shellfish/finfish aquaculture [76]. Currently, the main use of *Ulva* biomass as feed is for abalone, shrimp, and fish aquaculture [77,78,79,80,81]. Previous works in finfish aquaculture demonstrated the positive impact on fish growth of the inclusion of green seaweeds in their diet. For example, Hashim and Saat (1992) [82] showed that the incorporation of 5% *Ulva* spp. in the feed for Snakehead murrel resulted in an increase in growth rate, feed efficiency and feed consumption. The positive influence of including *Ulva* in fish feed has been demonstrated for other fish species, such as the Atlantic salmon, European sea bass and Nile tilapia [81,83,84]. 

*Ulva* biomass can be considered a good substitute feed for livestock because it contains more crude protein and minerals than traditional forages [85]. *Ulva* biomass can be incorporated into poultry diets, leading to a nutritional advantage [86]. Indeed, Abudabos et al. (2013) [38] demonstrated that replacing 3.0% of corn in the diet with *U. lactuca* for 21 days, had no negative effects on the measured production parameters of poultry, but improved carcass characteristics in terms of dressing and breast yield. 

Ulvans are the most abundant chemical compounds showing biological activity in *Ulva.* An ulvan extract of *U. compressa* has been reported to have antiviral activity, inhibiting virus propagation [87]. In U. rigida, antioxidant activity has been associated with its high polyphenolic content [88]. The crude extracts of two others *Ulva* species, *U. intestinalis* and *U. lactuca*, have also demonstrated antiprotozoal and antimycobacterial activity [89]. Sterols from *Ulva* have been reported to reduce blood cholesterol levels and were found to reduce excessive fat deposition in the heart [90]. Beyond *Ulva* biochemical compounds, the antimicrobial potential of *Ulva* epiphytic bacteria has also been reported in the species *U. rigida* [91]. Despite the value of these compounds, procedures for their large-scale extraction are still largely under development. Further advances in these methods are needed to support widespread usage and examination by the pharmaceutical industry [34].

This biomass is up to 50% carbohydrates, with no lignin, making it an excellent candidate for the production of bioethanol [53,92]. With the aim of decreasing reliance on fossil fuels, plant and algal biomass is considered a promising source of raw materials for biofuel production. Among the different biofuel types produced from macroalgae biomass are biodiesel, biogas, biomethane, hydrogen and bioethanol, with the latter being the most widely produced fossil fuel alternative from plant material [93,94]. The use of macroalgae biomass as a raw material for biofuel production offers interesting advantages over many land plant based biomasses. Among the benefits, the use of seaweeds avoids competition with food crops for arable land, many macroalgae do not require freshwater, which is increasingly scarce in many areas in the world, and often macroalgae cultivation does not require the supply of fertilisers, which are environmentally and economically costly [95]. Qarri and Israel (2020) [96] have also demonstrated that *Ulva* spp. dried biomass contained 16 to 22% of its dried biomass as TRS (total Reducing Sugars), and TRS showed a conversion rate of 30% to ethanol upon fermentation. The feasibility of producing *Ulva* feedstock in outdoor land-based cultivation for bioethanol production has been investigated and economic analyses have been conducted [96,97,98,99]. A long-term research program has shown that outdoor ponds of 1000 m^2^ can produce 10 tons of *Ulva* biomass (DW) per year which can generate about 730 L of ethanol [96,98]. However, if biofuel production from seaweed biomass has some potential, the use of this biomass for food/feed or extraction of high value compounds is probably economically and environmentally more promising. Hence, we suggest that biofuel production from seaweed biomass shall be beneficial in the frame of a zero-waste strategy, e.g., by producing biofuel after the previous extraction of valuable compounds from the biomass. 

Finally, the high growth rate of Ulva biomass, up to 30% per day, implies that *Ulva* spp. has a high nutrient assimilation potential. It makes them a suitable candidate for bioremediation processes which could eventually allow for offsetting the biomass production costs [2,100,101,102,103]. Moreover, because of their tolerance to different salinities, the use of *Ulva* spp. for the bioremediation of a wide range of wastewaters is possible [22,104]. Several studies have reported the nutrient removal efficiency of *Ulva* spp. with an assimilation rate for ammonium in the range of 50–90 μmol N g^−1^ DW h^−1^, the variation is explained by the species and growth conditions used [17,105,106,107,108]. Moreover, as mentioned before, *Ulva* species are known to rapidly accumulate high concentrations of heavy metals from its environment, which makes it suitable for heavy metal bioremediation [109,110]. *Ulva* spp. is also already used for bioremediation in shellfish aquaculture and several studies have demonstrated its effectiveness in the treatment of aquaculture effluents and bio-reactor sludge [15,17,18,19,22].

In recent years, coastal eutrophication associated with shellfish and finfish aquaculture has been a rising issue. In this context, integrated multitrophic aquaculture (IMTA), i.e., co-cultivation of marine livestock with primary producers, such as seaweeds, would mitigate the detrimental effects of aquaculture, enhancing its sustainability. The integration of *Ulva* into livestock monocultures, e.g., shrimp, urchin and abalone has led to several benefits, including a reduction in effluent nutrient loads released into the environment, a reduced need for commercial feed as *Ulva* spp. can be used as feed for those species (see above), and in some cases an increase in the economic value of the final product [19,22,111]. The global distribution of the genus *Ulva* also suggests that this approach may be applied worldwide [18,112]. However, species or even the strains which could be used need to be carefully selected. Indeed, it has been shown that species and strains respond differently in terms of growth performance, biomass composition and nutrient uptake in response to variations in environmental factors, such as nutrient source and concentration, salinity, or water temperature [107,113,114].

## 3. Strategies to Improve *Ulva* Biomass Yield and Composition

*Ulva* is a valuable marine resource, and its biomass can be used for a lot of purposes (see above). However, its biochemical composition varies significantly according to the strains, species, environmental conditions, and likely other factors, such as its associated microbiome [19,32,35,115,116]. Environmental growth conditions are thought to be the main factor influencing the composition of seaweed [117]. However, a recent study has shown that genetic variation can lead up to a five-fold variation in major compounds of *Ulva* biomass [23]. In addition, if the genotype (G) and environment (E), are components explaining the yield of a crop, their interaction (G*E), is also a major component, as it has been widely described for land crops [118,119]. Unfortunately, G*E has not yet been properly described in *Ulva* spp. In other words, *Ulva* spp. phenotypic variation is extensive, and the respective contributions of the genotype, environment and their interaction remain to be described. Another significant consideration for *Ulva* spp. phenotypic outcomes is the mutualism with microorganisms. *Ulva* can be considered a holobiont with its associated microbiome which is necessary for the proper development of the organs (rhyzoids and thalli) [120]. It is also likely that the microbiome is required for optimal vegetative growth [120]. A lot of research is still required, especially for large-scale cultivation where the biomass is usually outdoors and subject to fluctuating environmental conditions [121].

### 3.1. Environmental Conditions, a Focus on Salinity

Salinity is one of the most important factors influencing the distribution of the coastal green seaweed *Ulva* [122]. It is also probably the most important environmental factor to consider for the use of *Ulva* spp. in the bioremediation of coastal fresh/brackish wastewaters or in any other aquaculture application outside the sea. *Ulva* belongs to the intertidal zone and is a euryhaline genus that tolerates a wide range of salinities, from the hypohaline to hyperhaline zones [123]. Generally, intertidal seaweeds have a higher capacity to withstand changes in salinity than subtidal species. For example, the seaweeds present in the pools formed during the retreat of the sea can experience large variations in salinity during the day, ranging from 0.1 to 3.5 times that of seawater [124]. Importantly, *Ulva* spp. have the remarkable characteristic to include marine, brackish and freshwater species [123]. This makes this genus ideal for the investigation of the mechanisms involved in salinity tolerance and adaptations, as well as a source of diverse species which can be deployed for bioremediation, according to the salinity observed in the wastewater to be treated and the availability of water sources. 

Variations in salinity can cause osmotic, ionic and oxidative stresses, which have a strong effect on the cellular functions of photosynthetic organisms [125]. Under salinity stress, variations in osmolarity disturb cell turgor pressure, ion distribution and metabolic reactions, and often lead to an accumulation of reactive oxygen species (ROS). This accumulation of ROS is responsible for damage to protein complexes, membranes and other cellular components, thus affecting metabolism and growth, leading in extreme cases to cell death [122,126,127,128]. Such damage can result from either hypo or hyper-salinity treatments [129]. A number of publications document the impact of salinity on the growth rate and nitrate and phosphate uptake of *Ulva* species [114,130,131]. Disturbances in carbon and nitrogen metabolism due to changes in hypersaline conditions have also been described in the species *U. pertusa*, with an increase in the compatible solute proline [132]. In response to hyposalinity stresses, growth and physiological impacts of salinity have been largely documented in a number of *Ulva* species, such as *U. intestinalis*, *U. prolifera*, *U. linza*, *U. limnetica*, *U. lactuca* and *U. australis* [129,132,133,134,135,136,137,138]. A six-day exposure of *U. prolifera* to hyposaline conditions, from 30 ppt to 10 ppt, has a significant impact on growth rate and photosynthetic performance, decreasing growth rate by 65% [139]. Lu et al. (2006) [129] also found that after only 4 days of exposure of *Ulva fasciata* to hyposalinity (10 ppt), there was a reduction in maximum photosynthetic quantum efficiency (Fv/Fm) of 10%, which was proposed to be due to oxidative damage in chloroplasts [129]. 

Salinity is also known to affect the morphology of *Ulva* species [140,141,142,143]. Some *Ulva* species, such as *U. compressa* and *U. intestinalis* can be found with two distinct morphotypes, tubular and foliose thalli. Indeed, *U. compressa* is found as a monostromatic tubular morphotype in a saline/hypersaline environment and a distromatic foliose form in a low salinity environment, such as estuarine sites [141]. It is not clear whether these differences can be attributed to a direct effect of salinity or are an indirect effect of salinity associated variation in the microbiome [144]. No obligate foliose species have been recorded in freshwater ecosystems and tubular morphotypes are found in a broader range of salinities [3,145,146]. For example, *U. flexuosa* is the only *Ulva* species known to date which is able to grow from ultra-oligohaline to hyperhaline zones where salinity exceeds 50 PSU [4,142,145,147]. *U. torta* also shows a very wide range of salinity tolerance, from 1 to 36 PSU [123]. Valiela et al. (1997) [148] have hypothesised that those tubular cells have a better survival potential under low salinity conditions, e.g., a higher surface-volume ratio allowing for more rapid nutrient uptake compared to the foliose morphotype. In addition, an increase in the number of branches associated with a thallus in an aggregated form under low salinity has been reported for the species *U. prolifera* [6]. It was hypothesised that this may allow for better protection against increased turgor caused by lower salinity, as this new morphology would allow for the establishment of a more stable microenvironment around *Ulva* thalli. Contradictory observations were made in the distribution of *U. compressa*, which more frequently presents the tubular morphotype at high salinity and the foliose morphotype at low salinity [141,144]. Going further, Rybak et al. (2018) [123] hypothesised that an ancestral tubular morphotype carried tolerance and rapid adaptation mechanisms that are independent of morphotype, with these being lost among more recently diverged foliose and/or tubular species, but experimental evidence is unfortunately lacking. 

Recent gene expression studies identified candidate genes for involvement in tolerance to short-term low salinity conditions [125]. In one study, genes involved in photosynthesis and glycolysis were typically shown to be up-regulated in response to hypo-salinity stress [125]. An earlier study demonstrated the downregulation of many genes related to lipid metabolism, membrane and cell adhesion (51–93 genes) when *U. prolifera* and *U. linza* were cultured in fresh/brackish water compared to seawater [149]. The same study also identified some upregulated genes, encoding an ion transporter, a hydrolase and multiple heat shock proteins. Despite the insight that such comparative transcriptomics can offer, a more thorough understanding of the mechanisms of acclimatisation and tolerance to salinity variations is likely to require targeted strategies to identify genes involved in the process, such as via genome-wide association studies or QTL mapping. 

### 3.2. Microbiome

*Ulva* spp. depend on mutualistic bacteria for proper development and growth [1,150,151,152]. This dependence is not related to the presence of a single, defined bacterium, rather, it can be achieved by redundant partnerships and the details of these requirements are poorly described [153]. A useful study system for this dependency has been developed and termed “tripartite symbiosis”, where *Roseovarius* sp. MS2 and *Maribacter* sp. MS6 are sufficient to restore normal development in the *Ulva* host [154]. *Ulva* does not survive or grows at a very low rate with an undeveloped cell wall when deprived of its microbiome [120]. *Ulva*-associated bacteria also provide nutrient cycling and disease resistance for their host [155,156].

The change in environmental conditions during establishment in aquaculture settings often causes stress in the seaweed and changes in the associated microbiome [155,157]. *Ulva* adaptation to new environmental conditions can be considered to occur via changes in the metabolism of the seaweed depending on its genetic characteristics and changes in the bacterial community associated with *Ulva* that provides support through the production of algal growth and morphogenesis-promoting factors (AGMPFs) [158]. The composition of the microbiome associated with *Ulva* spp. is influenced by the geographical location as well as abiotic factors, such as temperature, salinity and nutrient concentrations [159,160]. Even if a core microbiome with the essential bacteria exists in macroalgae, Burke et al., 2010 [159] and Tujula et al. (2010) [161] have demonstrated that the composition of the microbiome changes both seasonally and geographically. Understanding this microbiome-*Ulva* complex is, therefore, essential given its importance for the adaptation of *Ulva* spp. to its environment, which will vary between aquaculture systems.

Many studies have examined the impact of growing conditions on the epiphytic microbiomes of seaweed [162,163]. In *Fucus vesiculosus*, an increase in salinity can cause a significant loss in bacterial community diversity [164]. Saha et al. (2020) [162] have shown that the epibacterial communities of an invasive red seaweed (*Agarophyton vermicullophylum*) changed significantly in terms of species richness and diversity according to the salinity. Concerning *Ulva* species, Tujula et al. (2010) [161] have shown that the microbiome associated with the species *Ulva australis* can vary considerably among the individuals collected from the same area and between different seasons. Califano et al. (2020) [157] have investigated the impact of wild *Ulva* transfer in a controlled environment (IMTA) on the composition of its microbiome and showed that the implementation of IMTA results in detectable changes in the epiphytic bacterial community. Another more recent study, focusing on the impact of one environmental factor, salinity, on the *Ulva* bacterial community has shown that the *Ulva*-associated microbiome is strongly structured by salinity [144]. Interestingly, the differences in bacterial communities at low and high salinity were quantitative rather than qualitative. These studies highlight that changes in bacterial communities are strongly environment dependent, which is an important consideration for the establishment of a new *Ulva* aquaculture farm [91,165,166].

To date, studies on associations between microbiota and conditions remain correlative, and only hypotheses can be made regarding the ability of bacteria to facilitate host adaptation to environmental factors. While studies have identified the bacteria required for *Ulva* development [1,150,151], studies identifying specific bacteria influencing the growth of mature thallus and the biochemical composition of the biomass are still lacking. To date, a limited number of studies have attempted to demonstrate that certain bacteria can promote *Ulva* growth [167,168] and can affect the biochemical composition of *Ulva* [116]. Further, examination of the molecular mechanisms driving *Ulva*: microbial interactions is still limited. For example, are there certain bacteria adapted to a specific environment that may be better than others for promoting *Ulva* growth? If they exist, such bacteria could be of critical importance to the optimisation of *Ulva* yields and biomass composition in aquaculture conditions. Thus, the use of different “cocktails” of bacteria could directly impact the biochemical content and the growth of *Ulva* [158,169]. Future studies should investigate the effect on *Ulva* phenotype of the microbiome:host genotype interactions, and the impact of environmental conditions on these interactions. For example, the exchange of resources and chemical signals from both host seaweed and epiphytic bacteria, and the impact of environmental conditions on these exchanges, should be documented. 

### 3.3. Natural Variation within the Genus Ulva spp.

Natural variation refers to changes in phenotype between individuals from the same species, which are explained by genetic differences. As a result, to assess the extent of natural variation within a species, individuals must be grown in the same environmental conditions in order to exclude changes in phenotypes due to the environment. Natural variation within *Ulva* species has been studied both for foliose and tubular species [23,170,171,172]. Lawton et al. (2013) [171] reported high levels of variation in the specific growth rate of the foliose thallus of *U. ohnoi*, with strains cultivated in the same location showing > two-fold variation in growth rates. Fort et al. (2019) [23] also reported extensive variation within *U. lacinulata* species, >four-fold, from 0.092 to 0.371 mg·mg^−1^·d^−1^. This variation was in fact as high as that observed between six different *Ulva* foliose species. Moreover, the authors reported a similar extent of natural variation for a large range of biochemical traits, e.g., starch content and protein content. Interestingly, Fort et al. (2020) [103] subsequently reported that for a given species, the *Ulva* strains originating from green tide areas have higher protein, pigments, lower starch content and higher growth rates than other samples, making green tide areas suitable places for the collection of strains for aquaculture if the biomass produced is destined to feed/food applications. Huo et al. (2013) [173] also identified several strains from a same species in greentides. 

Although natural variation has been identified as being very high within *Ulva* species, the associated genes are still unknown. A recent study has demonstrated the importance of intraspecific variation in mitochondrial genomes within the species *Ulva compressa* [174]. However, many previous studies of *Ulva* organellar genomes have shown very few differences within *Ulva* species, and high variation between species, suggesting that a large part of the natural variation within *Ulva* species is explained by nuclear encoded genes [43,172]. A recent study written by Fort et al. 2022 [11], details the genomic resources available in *Ulva*. Hence, nuclear DNA marker association studies, such as genome wide association studies or quantitative trait loci analyses, should be considered with growth and metabolite profiles to engineer, select or breed for improved yield and biomass characteristics in aquaculture.

However, before undertaking such targeted improvement strategies, significant productivity gains can already be achieved by simply screening this existing natural genetic variation to identify and isolate fast-growing strains with desirable characteristics. An important aspect of strain selection is to select strains in the environment they will be cultivated in afterward. The phenotype is dependent on the genotype (G), as well as the environment (E), and their interaction (G*E); hence, strain selection must be performed under environmental conditions as close as possible to those the strains will be cultivated in. To avoid the introduction of invasive strains, representing a threat to ecosystem balance and biodiversity and to ensure the preservation of the local genetic diversity, we suggest that such selection should be performed using local strains.

## 4. Conclusions

The global distribution, with wide environmental tolerance, high growth rate and nutrient uptake as well as a unique biochemical composition has made the green algal genus *Ulva* an attractive model for aquaculture and bioremediation. *Ulva* spp. biomass is becoming increasingly important economically, with many different industrial applications investigated, but the economic viability of large-scale cultivation needs further consideration. 

A “perfect programme” to obtain the best *Ulva* product is obviously very complex to define because of the many parameters which can influence the yield and quality of the biomass produced (Figure 1, “Phenotype”). Each aquaculture system has different growing conditions (“E”) that will have a direct impact on the final product obtained, but the impact of natural variation (“G”), microbiome composition (“M”), and the interaction between all these factors cannot be neglected. Three-way interactions between G, E and M may also be important [175], particularly in key developmental phases, such as substrate adhesion during colonization [176]. Furthermore, it is important to emphasise that the optimal growing conditions for biomass yield do not necessarily correspond exactly to the ideal growing conditions for obtaining a valuable final product for the desired application. 

The species within the genus *Ulva* possess a large diversity in environmental tolerances, necessitating the careful selection of a species for cultivation to achieve a desirable balance of biomass yield and biochemical composition. Therefore, further research should be conducted on improving the selection of strains according to the application, and to facilitate this work, the identification of the genes involved should be considered as they could be used, for example, as markers to assist the selection process. Moreover, those genes could provide the basis for genetic engineering to introduce novel traits and/or optimise metabolic throughput towards a desired biochemical composition. The creation of transgenic macroalgae has already proven to be successful in the genus *Ulva* and the progress of knowledge in this field seems to be promising [177,178]. However, the acceptance of these modified organisms remains questionable. Will the cultivation/commercialisation of an improved/engineered strain be accepted, even if it is generated from a local genotype? What will be the potential impacts of the spread of these modified genes on local biodiversity, as *Ulva* is already considered a highly invasive species worldwide? Moreover, for such targeted improvement strategies to succeed in *Ulva*, more genomic resources are required to empower genomic selection and molecular breeding. Fortunately, the extensive phenotypic diversity present among wild isolates means that large-scale selection programs, supported by marker-assisted selection, but not involving transgenic approaches, are likely to achieve significant improvements to yield, resilience and biomass quality [179]. 

While large-scale cultivation of *Ulva* spp. is still in its infancy, *Ulva* species represent a promising source of biomass with many exciting valorisation opportunities. Exploiting the valuable ecosystem services that *Ulva* can provide, such as in wastewater bioremediation, provides new avenues to increase the industrial competitiveness of *Ulva* cultivation.

## Figures and Tables

**Figure 1 biology-11-01593-f001:**
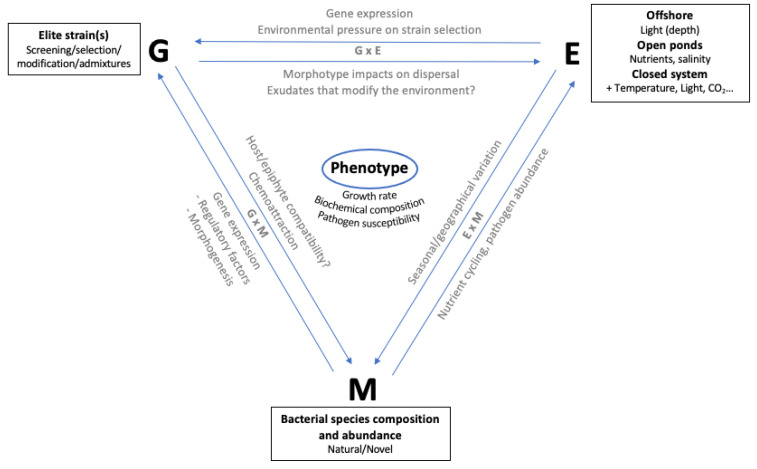
Description of the main factors controlling the phenotype. G: Genotype [123,151,152,177]. Elite or strains of interest can be selected by screening wild type strains and/or the creation of artificial populations. E: Environment [103,125,159,161]. M: Microbiome [155,156,169]. Variations in the impact of the microbiome on *Ulva* phenotype are expected via qualitative and qualitative changes in bacterial composition. Importantly, the interactions between those three factors are also expected to influence *Ulva* phenotype.

**Table 1 biology-11-01593-t001:** Biochemical composition of the macroalgae *Ulva* spp. and conventional feedstuffs (% Dry Weight).

	*Ulva* spp.	Soybean	Corn	Wheat
Proteins	9–29	37–43	10	9–19
Carbohydrates	41–50	20–30	74–85	61–84
Lipids	1–12	20	4	2
Ashes	14–52	6	1.2	1.5–2
Magnesium	2–5.2	0.12	0.13	0.14
Potassium	0.7–1.5	1.5	0.29	0.4
Calcium	0.8–6.2	0.3	0.007	0.04
Sodium	0.4–2.9	0.3	0.04	0.001

Sources: *Ulva* spp. [26,36,37,38]; Soybean [39,40,41]; Corn and Wheat [42,43,44,45].

**Table 2 biology-11-01593-t002:** Essential amino acids composition of the macroalgae *Ulva* spp. and conventional foodstuffs (g/100 g proteins).

	*Ulva* spp.	Soybean	Corn	Wheat
Phenylalanine	3.9–7.1	2.4	3.5	4.1
Leucine	4.6–6.9	7.3	8.8	5.9
Methionine	1.4–2.6	1.2–1.4	0.9	1.3
Lysine	3.5–4.5	6.4–6.5	1.8	2.9
Isoleucine	2.3–3.7	3.6	2.5	1.8
Valine	4.1–6.2	4.5	3.0	3.1
Threonine	3.1–6.9	4.0	2.0	2.9
Histidine	1.2–4.0	3.8–4.0	2.0	3.8

Sources: *Ulva* spp. (*U. lacinulata U. pertusa* and *U. lactuca*) [30,51]; Soybean [63]; Wheat [45]; Corn [64].

## Data Availability

Not applicable.

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
