# Peer review of "Applications of Ulva Biomass and Strategies to Improve Its Yield and Composition: A Perspective for Ulva Aquaculture"

_biology, 2022, doi:10.3390/biology11111593_

Round 1

Reviewer 1 Report

In this review, the authors describe biochemical content of Ulva, current industrial applications and strategies to improve Ulva biomass yield and composition. Furthermore, the authors elegantly summarize the text in their conclusions while highlighting interactions between Genotype (G), Environment (E) and microbiome (M). Although these topics are broad, the authors succeed in condensing a large body of available literature in concise, to-the-point paragraphs. I think this overview is timely and will be a valuable resource for researchers working on basic and applied Ulva research. 

I have some minor remarks:

The use of Ulva as a biofuel is discussed. The paragraph (3.3) could be strengthened by adding some numbers to demonstrate or dispute the viability of using Ulva biomass for biofuels. For example: how much bioethanol can be extracted from 100kg of Ulva? Is it necessary to cultivate Ulva in bioreactors, perhaps under mixotrophic conditions, or is biomass generated from other aquaculture systems (or green tide biomass) sufficient?

The authors focus on bioremediation of wastewater (paragraph 3.4). Different studies also describe the use of Ulva in bioremediation of heavy metals. This has implications, for example the high level of organic arsenic in seaweeds has raised concerns, which is linked to food and feed production (paragraph 3.1) and also biochemical composition (paragraph 2).

In the “Natural variation within the genus Ulva spp.” paragraph (4.3) I would suggest to include the study of Huo et al Limnol. Ocenogr. (2013) to further support that different genotypes of the same species can be present in green tide blooms to complement the authors own work on natural variation in Ulva. Also, Liu et al. Front. Mar. Sci. (2020) demonstrate that intraspecific variation in mitochondrial genomes exist in Ulva compressa; so I suggest to adapt Lines 433-435. Furthermore, it would be valuable to summarize the available genomic resources for Ulva (perhaps in a table) as these represent references for future association studies (Lines 432-438).

In this review, the authors present good arguments why Ulva is an attractive species to cultivate for many applications and give excellent suggestions on how yield can be improved. In particular, the natural variation in growth and biochemical composition represents an excellent resource to select for certain traits. It is however unclear if Ulva species lack certain metabolic pathways or other features that could be introduced by genetic engineering. Alternatively, do Ulva species have certain traits exist that should be selected against (e.g. heavy metal uptake)? Gene identification and marker-assisted breeding or genetic engineering is an interesting approach for targeted improvement of Ulva (Lines 440-442 and 476-480). I think it is worthwhile to include that Ulva is the only seaweed where both gain- and loss-of function mutants can be generated currently. At the same time, it is important to reflect on the commercial value of such strains: is there acceptance to grow an improved/engineered strain, even if they are generated from a local genotype (Line 447-449)? 

Line 277: use rhizoid instead of rhyzoid

Reviewer 2 Report

See attached file

Author Response

The responses to the comments of the reviewer are in the attached file

Reviewer 3 Report

The authors have assembled an excellent review of the use of Ulva biomass and strategies for improving the aquaculture of this under-utilised seaweed. The review is very well written and set out. It was a very nice read. I have no issue recommending this manuscript for publication after a few minor editorial and grammatical changes are made. I have included a PDF with specific changes suggested, but overall a lot of the changes come down to consistency of formatting, especially of the in-text citations. It would be a good idea for the authors to go over the in-text citation and bibliography carefully to ensure there is consistency throughout the manuscript. Additionally, the conclusion section felt out of place from the rest of the manuscript. I have suggested a couple of changes, but the whole conclusion could be reworked to better reflect the topics that were covered throughout the main body of the review.
Overall though, excellent work!

Round 2

Reviewer 2 Report

Comments to the Author

Biology (ISSN 2079-7737)

Applications of Ulva biomass and strategies to improve its yield and composition: a perspective for Ulva aquaculture (biology-1883345)

General comments:

Thanks to the authors to express their point of view. I would like to highlight that the authors have produced a well-written review and the effort and time poured into this piece of work is by all means obvious. However, I still feel the same about this review - that the authors have missed the opportunity to focus on one specific area. In addressing my previous comments, the authors have answered the question what this review should be themselves – a review to guide the management of Ulva for aquaculture. In my point of view, their review starts weak and finishes strong. The authors should focus their review on section 4 (Strategies to improve Ulva biomass yield and composition) as this is their strong point and sets this review apart from other published reviews and makes their work unique.

I still have an issue with section 2 (Ulva biochemical content) as this is not in depth and not any different to previous reviews out there (Bolton et al 2016 and Mantri et al 2020). Similarly, I feel that section 3 (applications) has a lot of overlap with previous reviews (Bolton et al 2016 and Mantri et al 2020) and doesn’t offer much new insights. 

My recommendation is to reject this review, but invite the authors to resubmit their review after substantially rewriting their work. Please find more detailed comments below.

Specific comments:

1. Introduction:

As this review has a focus on aquaculture, I recommend to delete the second paragraph, which focuses on green tides. Removing this paragraph will also enhance the flow of the introduction.

2. Ulva biochemical content:

I recommend removing the subheadings and tables from this section. The relevant information of the table can be added to the text. I understand that the authors are keen to keep the biochemical content as this sets the context of why we should bother to grow Ulva, but as the authors don’t discuss the specific biochemistry (e.g. amino acid profile, specific magnesium content, etc) in any other section of the review, this seems a little redundant and also takes away from the question of the review. Check out the reviews by Bolton et al 2016 and Mantri et al 2020 how they have incorporated a section of the biochemical profile without making it a major part of their review.

3. Ulva spp. application

I understand the authors want to highlight all the wonderful (theoretical) applications of Ulva. However, this section also takes away from the focus of the review “to improve the understanding and management of Ulva biomass in aquaculture”. I agree that it’s important to provide a context why we want to grow Ulva and biomass applications are the major part of this story. In my opinion, I would have a short paragraph highlighting the most important applications (food/feed, high value (nutraceutical) and bioremediation) and then move on. Most of the authors text under the “Food and feed” section is highlighting the constrains due to legislative regulations in the Western world and this is an interesting point of view. The authors could focus on this area. However, I recommend removing the biofuel section and advise the authors to read “The hype, fantasies and realities of aquaculture development globally and in its new geographies” by Costa-Pierce and Chopin. I would also like to add that the world is moving away from fuels in general to renewables and electricity. As the authors point out in their review, Ulva is a great food and feed source and can contribute to securing food for the world, while converting it into biofuel does not seem to do Ulva justice.  

4. Strategies to improve Ulva biomass yield and composition for bioremediation

Why is this section limited to bioremediation? It makes perfect sense to use Ulva for bioremediation, but by broadening this section to cultivation in general (which also includes for bioremediation purposes) the authors make their review relevant to a broader field (change heading to “Strategies to improve Ulva biomass yield and composition in aquaculture”) . The authors cite a few studies in this section without letting the reader know the outcome of the studies (e.g. ln 313-315, ln 343, ln 345).
I enjoyed reading the salinity section and the focus on how salinity affects metabolic processes and morphology is of interest. However, the impact of salinity on the yield and composition is very limited in this section, although that is the focus of the review and this particular section (based on the heading). The authors have focused on this well in section 4.3 and a similar narrow focus would benefit the salinity section, as well as the microbiome section (which has also very limited information on the effects on biochemistry and growth of Ulva). I understand that the area of microbiome is rather new and that literature is very limited in this area, but please highlight how changes in microbiome affect these two factors (e.g. ln 424-427: what sort of changes did this result in? Lower growth? Changed composition?) as less than 25% of this section is about yield and composition of Ulva.

Author Response

The answer to reviewer comments is in attached file
